# Adherence and Effect of Home-Based Rehabilitation with Telemonitoring Support in Patients with Chronic Non-Specific Low Back Pain: A Pilot Study

**DOI:** 10.3390/ijerph20021504

**Published:** 2023-01-13

**Authors:** Peter Krkoska, Daniela Vlazna, Michaela Sladeckova, Jitka Minarikova, Tamara Barusova, Ladislav Batalik, Filip Dosbaba, Stanislav Vohanka, Blanka Adamova

**Affiliations:** 1Department of Neurology, Center for Neuromuscular Diseases (Associated National Center in the European Reference Network ERN EURO-NMD), University Hospital Brno, 625 00 Brno, Czech Republic; 2Faculty of Medicine, Masaryk University, 625 00 Brno, Czech Republic; 3Department of Rehabilitation, University Hospital Brno, 625 00 Brno, Czech Republic; 4Department of Public Health, Faculty of Medicine, Masaryk University, 625 00 Brno, Czech Republic; 5Department of Rehabilitation and Sports Medicine, Second Medical Faculty, Charles University and University Hospital Motol, 150 06 Prague, Czech Republic; 6Institute of Biostatistics and Analysis Ltd., 602 00 Brno, Czech Republic

**Keywords:** low back pain, paraspinal muscles, muscle strength, muscular endurance, pain, disability, adherence, telemonitoring, home-based rehabilitation

## Abstract

Home-based exercises have been on the rise recently. This pilot study aimed to assess the adherence and effect of a home-based rehabilitation programme using telemonitoring in patients with chronic non-specific low back pain (CNLBP). Twenty-seven patients with CNLBP were enrolled in the study, each of whom underwent a neurological assessment, including patient-oriented measures and a functional assessment—a battery of tests that comprehensively evaluated trunk muscle function. The rehabilitation programme lasted 18 weeks and included daily home-based exercises. A mobile application or an exercise diary was used to monitor compliance. Adherence to the programme was excellent for both the diary and mobile application groups, with 82.3% in the diary group exercising at least once a day and 72.9% twice a day, and 94.8% in the mobile application group exercising at least once a day and 86.6% twice a day. Both patient-oriented and functional outcomes improved significantly; however, the relative changes of the parameters in these two groups did not correlate, which supports the idea that trunk muscle function does not directly relate to patient complaints and that CNLBP is a multifactorial issue. This model of rehabilitation programme should be used in clinical practice, as its adherence and effectiveness seem noticeable.

## 1. Introduction

Non-specific low back pain (NLBP) is a very frequent condition and it is the most common type of low back pain (LBP). It is reported as a major health and socioeconomic problem associated with work absenteeism, disability, and high costs for patients and society [1,2]. In most cases, the source of pain is not established, and thus NLBP is defined as a pain that is not attributable to a recognisable, known, and specific pathology (e.g., tumour, osteoporosis, infection, fracture, structural deformity), radicular syndrome, or cauda equina syndrome [2]. LBP is considered to be chronic (CLBP) when it persists for 12 weeks or longer. Although there is limited scientific evidence, the best estimates suggest that the prevalence of chronic NLBP (CNLBP) is about 23% [2].

One proposed mechanism for CNLBP is a lack of stability of the spine due to dysfunction of the deep spinal stabilisation system, which consists of the lumbar paraspinal muscles, abdominal muscles, diaphragm, and pelvic floor muscles. The dysfunction of this muscle system over time may lead to chronic back pain [3]. Proper function of the lumbar paraspinal muscles, particularly the erector spinae muscle and the lumbar multifidus muscle, which are part of the lumbar extensor muscle system, is especially important for spinal protection. The lumbar paraspinal muscles play an essential role in stabilising the lumbar spine and initiating and controlling all movements of the lumbar vertebral column [4,5]. At the same time, the lumbar extensor muscles protect the underlying osteoligamentous spinal structures [6]. There is evidence of an association between CNLBP and decreased strength, muscular endurance, atrophy, and excessive fatigability of the lumbar extensors [7,8,9,10,11,12].

Management of chronic low back pain (CLBP) is often multidisciplinary, involving a combination of treatments, including therapeutic exercises [13]. Consensus regarding evidence-based treatment recommendations for patients with LBP from the recent European clinical practice guidelines identified a wide range of predominantly non-pharmacological treatment options. There have been strong recommendations for advice and education, remaining active, exercise programmes/therapy, and return-to-work programmes, as well as moderate recommendations for manual therapy combined with other treatment, group exercise programmes and back schools, psychological therapy combined with other treatments, and work-based interventions [14].

Exercise programmes may include different types of exercises and activities. Among the most commonly used exercise interventions are core stability exercises that are focused on the activation of the deep trunk muscles and target the restoration of control and coordination of these muscles [1,15]. Core stability exercises aim to reduce pain and disability in CLBP, to increase spinal stability and neuromuscular control, and to prevent the progression of shear force causing injury to the lumbar spine [13,16,17]. In the literature, these exercises are often referred to as motor control exercises [1]. A recent review proved that core stability exercises are more effective than rest or no/minimal intervention, that the combination with other types of exercise for CLBP has greater efficacy, and that patient compliance is crucial in determining the efficacy of the intervention [13].

Since a greater prevalence of diaphragm fatigue has been proven in CLBP individuals as compared with healthy controls, and diaphragm fatigue results in a lack of active spinal control, it is advisable to include respiratory training in the exercise programme for patients with CLBP [18,19].

Back school is also often used as a part of the treatment in patients with LBP. It is an educational programme that includes information on the anatomy of the back, biomechanics, optimal posture, ergonomics, and back exercises, and was designed to reduce back pain and prevent recurrences of LBP episodes [20,21]. Theoretical information could help patients understand their condition and learn how to modify their behaviour with regard to LBP.

Home-based rehabilitation is defined as a series of exercises that patients complete at home for therapeutic gains or to improve physical capacity. It is often associated with the involvement of telerehabilitation and telemonitoring, which use telecommunication technologies to control or monitor remote rehabilitation [22,23]. Telerehabilitation has been recently used in different fields and for various medical conditions (musculoskeletal, neurological, respiratory, cardiovascular, etc.) [22]. Webpages, phone calls, teleconference software, and messaging services are telecommunications technologies, or platforms implemented to deliver the rehabilitation. Virtual reality, understood as incorporating remote assistance from a therapist, is also used [22]. Home-based exercises are especially valuable because they require fewer resources and less time from health institutions and health practitioners [24,25]. Home-based exercises have been on the rise recently, spurred significantly by the COVID-19 pandemic. This enabled rehabilitation to continue even when elective care was restricted due to the pandemic, and it is also a promising way to mitigate the lack of available exercise centres [24]. Home exercises should be designed to be practical, accessible, and feasible.

The hypothesis of this study was that adherence and satisfaction with a home-based rehabilitation programme (HBRP) in patients with CNLBP would be favourable and that after completing this programme, both patient-oriented parameters and the functional parameters of trunk muscles would improve. The aim of this study was to assess the applicability of the HBRP with telemonitoring support and to evaluate adherence to exercise, patient satisfaction, and the effect of this programme.

## 2. Materials and Methods

This was a prospective interventional study. The local institutional medical research ethics committee approved the study protocol (agreement number 02-120220/EK), and written informed consent was obtained from all participants.

### 2.1. Participants

Most of the patients were referred by collaborating outpatient neurologists between May 2020 and January 2022.

Inclusion criteria were: age between 18 and 70 years and CNLBP (pain localised in the lumbar spine area and without radiation below the knee; pain duration over 12 weeks). Exclusion criteria were: previous lumbar spine surgery, presence of lumbosacral radicular pain in the medical history with residual signs of nerve root dysfunction in clinical neurological examination and manual muscle testing of the lower extremities, presence of myopathy, previous vertebral fracture, spine infection or tumour of the lumbar spine, comorbid conditions affecting the overall mobility of the patient, confirmed pregnancy, and significant impairment of cognitive functions. Discontinuation of analgesics was not required at study entry, but their use was monitored during the study.

### 2.2. Procedures

At the beginning of the trial, each patient underwent a detailed neurological clinical evaluation, including medical history and patient-oriented measures, performed by an experienced neurologist and a functional assessment of the trunk muscles by two physiotherapists. The intervention (HBRP) itself was led by two other physiotherapists and lasted 18 weeks. After this period of time, the assessment (neurological and functional) was repeated.

#### 2.2.1. Neurological Clinical Evaluation and Patient-Oriented Outcomes

Each subject was interviewed and examined by an experienced neurologist and the medical history was taken to evaluate the presence of exclusion criteria and comorbidities and the current intake of analgesics or muscle relaxants. Subjects underwent a comprehensive neurological examination. LBP was deeply analysed in terms of the intensity. Pain intensity was quantified using an 11-step numerical rating scale (NRS: 0–10). Current pain intensity together with average and maximum pain intensity in the previous 4 weeks were recorded. Disability in relation to LBP was evaluated using the Oswestry Disability Index (ODI) [26] and the Roland–Morris Disability Questionnaire (RMQ) [27].

After the 18-week rehabilitation programme had been completed, patient satisfaction with the rehabilitation programme and changes in patient beliefs (whether the patient believed that the rehabilitation programme helped with their LBP) were assessed using a semiquantitative scale with five levels (fully satisfied/rather satisfied/neither satisfied nor dissatisfied/rather dissatisfied/fully dissatisfied).

#### 2.2.2. Functional Assessment of Trunk Muscles

To assess trunk muscle function, a battery of simple tests was used, including strength and endurance measurements specific for core muscles with an emphasis on examining the back extensors. Maximal isometric lower back extensor strength was examined using a handheld dynamometer MicroFET 2 (Hoggan Scientific, LLC., Salt Lake City, UT, USA) in three postural positions (prone, sitting, and standing). To evaluate trunk and hip extensor endurance, the Biering–Sørensen test was used. The rest of the tests were more of a challenge for other muscle groups of the core trunk muscle system, which is crucial to trunk stability. These tests included measuring respiratory muscle strength—maximal inspiratory muscle strength (maximum inspiratory pressure, MIP) and maximal expiratory muscle strength (maximum expiratory pressure, MEP)—using the microRPM (Micro Medical, Kent, UK) electronic pressure gauge. Other tests challenging core muscle endurance were the prone-plank test for abdominal core muscles and the side-bridge test (on both sides) for lateral core muscles. The methodology is described in detail in a previous work by some of the authors of this study [28].

#### 2.2.3. Home-Based Rehabilitation Programme

The comprehensive rehabilitation programme (home-based exercise) was focused on activating the deep trunk muscles, on improving their coordination, and on strengthening them. For this purpose, a combination of several rehabilitation methods recommended in the treatment of LBP was used. This combination included back school, core stability exercises, respiratory training, and stretch exercises focused on the low back area.

During the entire rehabilitation programme, which lasted 18 weeks, each patient underwent seven regular check-up visits with a physiotherapist, during which the patient was introduced to the exercises to be performed at home and/or the correct performance of the exercises was checked. For the purpose of this study, three sets of exercises with gradually increasing difficulty levels were created, including the basics of back school, three core stability exercises, two stretching exercises and an exercise as respiratory training (Appendix A). The baseline visit at the physiotherapist, at which every patient obtained the first set of exercises, was followed by another visit after two weeks that enabled patients to ask questions in case of doubts and physiotherapists to check the actual patient performance of exercises. Four weeks later, there was a check-up visit with a physiotherapist, during which it was decided whether the particular patient was prepared to switch to a more difficult set of exercises or should stay longer at the same level. If the patient was not able to follow the expected amount of exercise in the optimal manner, they would be asked to continue on the same level for a while. The difficulty level of the exercise increased either in the sense of the postural position or in the number of repetitions or holding time, so that motor learning would be stimulated all of the time. This model of check-up consultations after two weeks and the intensity switch after four weeks was repeated two more times (Figure 1). The patient was asked to do the home exercises daily during the 18-week rehabilitation programme and was advised to exercise 15 min twice a day. Instruction on the rehabilitation programme was accompanied by visual support that the patient obtained as a booklet with pictures of exercises and precise descriptions. A total of four booklets were created. Three booklets were intended for the 18-week rehabilitation programme (each booklet covered six weeks). The fourth booklet, which the patient received at the last check-up visit, contained a summary of the exercises. At the last check-up visit, the necessity of regular exercise to prevent the recurrence of back pain was explained to the patient, who was advised to continue with the exercises as instructed in the fourth booklet.

### 2.3. Compliance and Telemonitoring

Telemonitoring was used to monitor patient compliance using a special mobile application adapted for this purpose. In this application, the patient was asked to record the frequency of exercise (whether they had done the exercise and how many times a day). With telemonitoring, investigators were able to contact the patient immediately via email or phone if it seemed that the patient was not following the programme and then discuss the potential problem. However, if the patient did not want/was not able to use this modern technology, there was an option to use an exercise diary in which they recorded information about the frequency of exercise. The diary was handed in at every check-up visit, and any problems that arose were discussed in person. The mobile application and exercise diary served as feedback for patients and investigators.

### 2.4. Statistical Analysis

Standard descriptive statistics were applied in the analysis. Continuous variables were described in median, minimum, and maximum. Categorical variables were characterised by absolute and relative frequencies.

The difference in adherence to HBRP between the subgroups was calculated using the Mann–Whitney test. The Wilcoxon signed-rank test was used to analyse the statistical significance of continuous data regarding the differences in functional and patient-oriented outcomes prior to and following the HBRP. The statistical significance of categorical data differences was calculated from the contingency table using Pearson’s chi-squared test with continuity correction and Fisher’s exact test.

The relative changes in particular parameters (patient-oriented outcomes and functional outcomes) were calculated as the ratio between the difference in the value of a specific parameter after and before the rehabilitation and its value prior to the rehabilitation.

Correlations of relative changes in particular parameters after the rehabilitation were computed using Spearman’s correlation coefficient (rs) and its corresponding *p*-value.

Statistical significance was set at *p* < 0.05. The analysis was performed in the software IBM SPSS Statistics version 28.0.0.0 (190).

## 3. Results

### 3.1. Patient Characteristics

Twenty-seven patients (15 men and 12 women) with CNLBP were included in the study. Figure 2 summarises subject recruitment. The basic characteristics of these patients are presented in Table 1. Median age was 49.0 (range 26–68) years. The median duration of CLBP was 96 weeks. Analgesic medication was used in 55.6% of patients; 14.8% of patients were taking central muscle relaxants.

### 3.2. Adherence to a Home-Based Rehabilitation Programme and Patient Satisfaction

Of the 27 patients who started the HBRP, a total of 25 (92.6%) patients were able to complete the programme. At the beginning of the HBRP, 5 patients (20%) used an exercise diary and 20 patients (80%) recorded the exercise using the mobile application. During the HBRP, 2 patients switched from the mobile application to the diary. Thus, in the end, 7 patients used a diary, and 18 patients used the mobile application. All 25 patients completed three booklets. The adherence of patients who completed the HBRP is analysed in more detail in Table 2. Patients using exercise diaries logged 100% of their records. Patients with the mobile application logged 97.1% of their records. Statistically significantly lower adherence to the HBRP (exercising at least once a day and exercising twice a day) was observed in the patients using the exercise diary than in the patients with the mobile application, as shown in Table 2. The two subgroups were significantly different (*p* < 0.001) in the percentage of days they did not exercise at all (17.7% with an exercise diary versus 5.2% of patients with the mobile application).

Patient satisfaction with the rehabilitation programme and changes in patient beliefs are summarised in Table 3. All 25 patients were satisfied with the HBRP (22 of them fully satisfied and 3 rather satisfied). Twenty-four patients (96%) believed that this rehabilitation programme helped them with problems associated with their lumbar spine.

### 3.3. Changes in Pain, Disability, and Functional Tests after the HBRP

There was a significant reduction in average pain, current pain, and maximum pain after completing the rehabilitation programme (Table 4). There was also a significant decrease in patient disability as expressed by the ODI and the RMQ (Table 4). A significant reduction in the use of analgesic medication was observed: 13 patients used analgesics before the HBRP started, and only 3 patients continued to use analgesics after the HBRP ended (Table 5). There was a highly statistically significant improvement in all functional parameters of the trunk muscles in patients after completing the rehabilitation programme. There were improvements in maximal isometric lower back extensor strength assessed in all three positions, in respiratory muscle strength, and in all trunk muscle endurance tests (Table 6).

### 3.4. Correlation of Relative Changes in Particular Parameters after the HBRP

The correlations of relative changes in particular parameters after the rehabilitation programme are shown in Figure 3. Relative change in average pain intensity statistically significantly correlated with relative change in maximum pain (rs = 0.67, *p* < 0.001) and with relative change in current pain (rs = 0.46, *p* = 0.02). The relative change in maximal isometric lower back extensor strength in a sitting position correlated with relative change in this strength measured in a standing position (rs = 0.64, *p* = 0.001) and with relative change in prone-plank endurance (rs = 0.58, *p* = 0.003). There were significant correlations between the relative changes in the particular endurance tests (i.e., Biering–Sørensen test, prone-plank test, and side-bridge test bilaterally). The relative change in maximal inspiratory muscle strength correlated with relative change in maximal expiratory muscle strength (rs = 0.41, *p* = 0.05) and with relative change in maximal isometric lower back extensor strength in a sitting position (rs = 0.59, *p* = 0.002), with relative change in prone-plank endurance (rs = 0.63, *p* = 0.001), and with relative changes in side-bridge endurance bilaterally (rs = 0.43, *p* = 0.03, resp. rs = 0.58, *p* = 0.003). Correlations between relative changes of patient-oriented outcomes (pain, disability) and relative changes of functional outcomes (muscle strength and endurance) were not proven to be significant.

## 4. Discussion

This study focused on analysing the applicability of a comprehensive HBRP using telemonitoring in patients with CNLBP. The adherence to training, patient satisfaction, and the effect of rehabilitation were evaluated.

The hypothesis of this study is proven. The results suggest that home-based rehabilitation may be applicable in clinical practice, with patients having very good adherence to the exercise programme and reporting satisfaction with the programme. Patients who used the mobile application showed slightly better adherence to exercise than patients who recorded their activity in a diary. The effect of rehabilitation on the clinical condition of the patient, including functional parameters of the trunk muscles, was also very positive. However, relative changes in particular parameters (patient-oriented and functional) after rehabilitation correlated with each other only to a limited extent, supporting the idea that trunk muscle function does not directly reflect patient complaints and that CNLBP is a multifactorial issue. To assess the effects of a treatment programme in patients with CNLBP, it is appropriate to use more than one parameter, including both patient-oriented parameters (such as pain, disability, use of analgesics) and objective functional parameters (such as strength and endurance of trunk muscles).

### 4.1. Adherence

There is no current ‘gold standard’ for measuring patient adherence to home-based exercise. A study by Peek et al. [29] compared two methods of reporting adherence to a rehabilitation programme for CLBP, by patient report and physiotherapist perception, and presented their observational data. The study found poor agreement between patient and physiotherapist self-reported measures. In the current study, each participant had the choice of using either a mobile application or an exercise diary, and they were allowed to switch from one to another, according to their preferences. Most participants (80%) chose to use a mobile application, and only two of them switched (to the diary) in the course of the study. Participants who used a diary kept a daily record of their exercise; those who used the mobile application reached almost the same frequency. Significantly higher adherence to the HBRP was observed in patients with the mobile application than in the patients with an exercise diary. This may be due to the attractiveness of using modern technologies, or to the feeling of being constantly watched. There were two incidents in which a patient did not enter any record into the application for three days in a row; clinicians were able to contact the patient and solve the problem effectively. The adherence was very high for both the diary and the mobile application groups, with patients exercising at least once a day 82.3% (diary) and 94.8% (mobile application) and exercising twice a day 72.9% (diary) and 86.6% (mobile application), in comparison to other studies, in which adherence mostly varied from 70% to 80% [29,30,31,32]. Here, a task that was hard to accomplish (to exercise twice a day) was used, so at least half of the task would be fulfilled (to exercise once a day). This strategy seemed to be very effective, as participants using the mobile application adhered to the programme, in terms of exercising at least once a day, almost fully. This is in consensus with authors, who have declared that fractionalisation of an exercise into multiple bouts spread across the day may produce greater benefits and enable greater adherence [33].

Medina-Mirapeix et al. focused on other possible aspects of adherence than the studies mentioned above [34]. They aimed to determine whether patients with neck or low back pain had different rates of adherence to exercise components in terms of frequency per week and duration per session when prescribed a home exercise programme, and to identify if adherence to both exercise components had distinct predictive factors. They found that adherence to duration per session was more probable than adherence to frequency per week. According to their findings, frequency adherence was more probable if patients received clarification of their doubts, and duration adherence was more probable if they were supervised while learning the exercises [34].

Another study showed negative results for a home exercise programme: 57% of patients did not return for the control evaluation [35]. This is in contrast to the current study, where all but two participants were able to complete the programme. The two dropouts were for personal issues unrelated to the study structure. In consensus with these authors, it may be concluded that clinicians should be aware of the patient adherence level when recommending home-based exercises and should also realise that exercises might be performed inaccurately in an unsupervised environment, so even home-based exercise should include regular check-up visits.

In a qualitative study discussing factors that could improve adherence to home-based rehabilitation programmes, the biggest roles were played by providing clinical knowledge, promoting feedback during exercise instruction, giving reminders, and monitoring results and adherence to exercises [36]. All of these components were covered in the current study. Other issues revealed as important in the work above were taken into account in our clinical trial, such as how the prescribed exercises were designed, the degree of difficulty of the exercises, and how the programme was delivered by the care provider.

### 4.2. Satisfaction and Beliefs

In this study, all patients were satisfied with the programme (22 fully satisfied, 3 rather satisfied), and 96% of the participants believed that this rehabilitation programme helped them with problems associated with their lumbar spine. However, those data were taken at the end of the programme. Studies focused on the long-term effect of their programmes suggested that the satisfaction with the programme may decrease over time and may change into non-positive appraisals, even though the satisfaction right after the end of the trial was rather positive [32].

### 4.3. Changes in Pain, Disability, and Functional Tests after the HBRP

To interpret changed scores for pain and functional status in LBP, there is an international consensus regarding the minimal important change proposed by values: 2 for the NRS (0–10), 5 for the RMQ (0–24), and 10 for the ODI (0–100). When the baseline score is taken into account, a 30% improvement should be considered a useful threshold for identifying clinically meaningful improvement on each of these measures [37]. This study followed the recommendation and showed a significant reduction in pain and a decrease in patient disability (ODI and RMQ). At the same time, it proved significant improvement in all functional parameters of the trunk muscles. We consider the fact that the number of patients taking analgesics dropped significantly, from 52% to 12%, after completing the HBRP to be very important.

A recent meta-analysis proved that home-based exercise training improved pain intensity and functional limitation parameters in LBP [24]. However, the analysis pointed out that this type of programme requires a high level of patient motivation, regular supervision, and patient evaluation; the current study tried to incorporate these requirements [24,32]. In contrast to the model of the current study, which was home-based including regular everyday exercise for 18 weeks, another study claimed that one lumbar extension training session per week performed in a medical setting is sufficient for strength gains and pain reduction in CLBP patients [38]. More studies are needed comparing different exercise intensities in terms of frequency, and comparing home-based rehabilitation with conventional rehabilitation in a medical setting to estimate and compare the size of the effect of each programme. One such study showed that a significant improvement (relative to baseline) in pain, trunk flexibility, muscle endurance, and psychological and functional measures was observed in both study groups (home-based and conventional treatment) but with a significantly greater difference in the group performing home-based rehabilitation [32]. Further, they found that the home-based group maintained the improvement in all parameters at a one-year follow-up visit. However, the conventional rehabilitation group only maintained their disability status. The current study did not follow the long-term effect of the programme. On the other hand, this is a pilot study providing a basis for further works comparing HBRP with standard rehabilitation and following the long-term effect of both.

As mentioned above, this study observed a significant improvement in all the assessed functional parameters of the trunk muscles in both strength and endurance. In addition, some studies suggested that besides improvement in functional parameters, exercise programmes may have an effect in increasing the cross-sectional area and improving the density of paraspinal muscles in patients with LBP [39,40].

### 4.4. Correlations in Changes of Particular Parameters

To our knowledge, few studies have discussed the relationship between changes in functional outcomes and patient-reported outcomes following lower back exercise. Nonetheless, some addressed this question and some even suggested that there may be a relationship between improvements in functional and patient-oriented outcomes. Steele et al. performed a secondary analysis that showed relatively consistent significant relationships between increases in isolated lumbar extension (ILEX) strength and reductions in both the visual analogue scale and the ODI after ILEX resistance training [41]. It is interesting to consider why a change in ILEX strength might be uniquely related to patient-oriented outcomes, whereas other functional measures are not. This may be related to the specific role that deconditioning of the lumbar extensor musculature might play in the initiation and development of CLBP [11,12]. The current study did not confirm these findings; however, ILEX resistance training was not the basis of the home-based exercise. The exercise programme used in this study focused on the activation of the deep trunk muscles. It targeted the restoration of control and coordination of these muscles, progressing to more complex and functional tasks integrating the activation of deep and global trunk muscles, making it a motor control exercise, as described in other studies [1]. No correlations between relative changes in functional and patient-oriented outcomes were found. This rules out the suspicion that the functional measures used in this study may be measures of pain-related behaviour, which is always a question while choosing functional measures [42]. The findings of this study support the idea that trunk muscle function does not directly reflect patient complaints; CNLBP is a multifactorial issue. To evaluate the effect of any intervention on CNLBP, both patient-oriented and functional outcomes should be considered.

On the other hand, relative changes in particular patient-oriented outcomes correlated with one another, especially pain intensity (average, maximum, and current). In functional outcomes, the relative change in maximal isometric lower back extensor strength in a sitting position correlated with the relative change in this strength measured in a standing position, supporting the construct validity of the measurements used. Moreover, relative changes in maximal isometric lower back extensor strength measured in a sitting position correlated with relative change in prone-plank endurance. Relative changes in all endurance tests correlated with one another. Furthermore, relative changes in inspiratory muscle strength correlated with relative changes in some strength and endurance tests. This supports our presupposition that the chosen exercise had the potential to increase the activation of the deep muscles of the trunk and improve the control and coordination of these muscles in terms of strength and endurance.

### 4.5. Limitations of the Study

The study has three main limitations: first, a relatively small number of patients; second, no randomisation and no control group; and third, no data about long-term adherence. However, this was a pilot study, the main aim of which was to evaluate the applicability and usefulness of a designed home-based comprehensive rehabilitation programme in patients with CNLBP. The results of this study served as a basis for another study that has already been initiated and which is randomised, comparing home-based rehabilitation with conventional rehabilitation in a medical facility. The strength of this study is the use of a wide range of parameters, not only patient-oriented outcomes (such as pain, disability, and change in the use of analgesics) but also a detailed assessment of trunk muscle function, to evaluate the effect of the rehabilitation programme. Another positive factor of the study is the use of modern approaches in rehabilitation, namely home-based rehabilitation and telemonitoring. For future studies, it would be useful to enhance the mobile application function in terms of data clarity for caregivers, so it would be easier for them to interpret data and immediately recognise a suspicious absence of record-keeping.

## 5. Conclusions

Adherence and patient satisfaction with the HBRP used in this study were favourable and exceeded our expectations. These were probably supported by the regular supervision (by means of telemonitoring and an exercise diary) and regular check-up visits to increase motivation. The use of telemonitoring via a mobile application likely further increased adherence to exercise. A notable effect of the comprehensive HBRP on patient clinical condition was demonstrated, including improved trunk muscle strength and endurance.

We believe this type of rehabilitation programme could be used in everyday clinical practice in patients with CNLBP. Further study is needed to prove its applicability in clinical routine.

## Figures and Tables

**Figure 1 ijerph-20-01504-f001:**
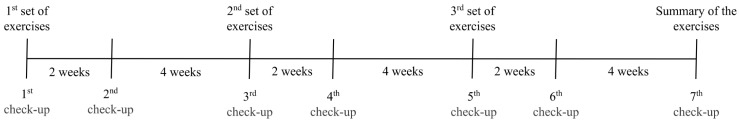
Timeline of the rehabilitation programme.

**Figure 2 ijerph-20-01504-f002:**
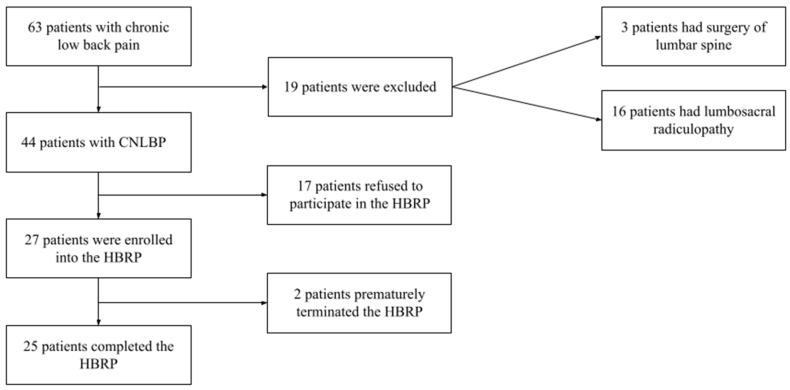
Study flowchart, including subject recruitment.

**Figure 3 ijerph-20-01504-f003:**
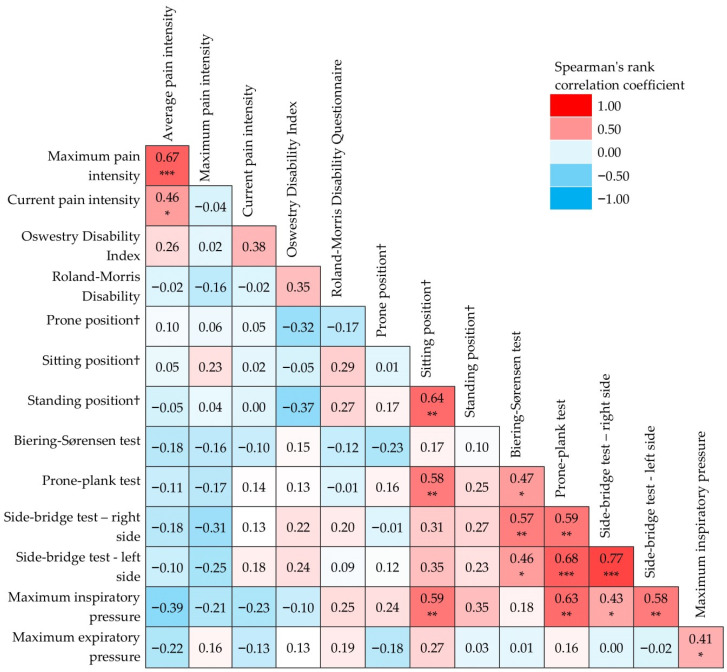
Correlation of relative changes in particular parameters after the HBRP. Significance of correlations is expressed through asterisks at levels: * *p* < 0.05, ** *p* < 0.01 and *** *p* < 0.001. ^†^ Maximal isometric lower back extensor strength in different positions.

**Table 1 ijerph-20-01504-t001:** Characteristics of subjects.

Variable		Patients Started HBRP (n = 27)	Patients Completed HBRP (n = 25)
Sex	Female [n (%)]	12 (44.4)	10 (40.0)
	Male [n (%)]	15 (55.6)	15 (60.0)
Age [years]		49.0 (26; 68)	49.0 (26; 68)
Height [cm]		175.0 (160.0; 188.0)	176.0 (160.0; 188.0)
Weight [kg]		79.0 (56.0; 118.0)	80.0 (56.0; 118.0)
BMI [kg/m^2^]		25.8 (19.4; 35.2)	25.8 (19.4; 35.2)
Education	Primary education [n (%)]	10 (37.1)	9 (36.0)
	Secondary education [n (%)]	8 (29.6)	8 (32.0)
	Tertiary education [n (%)]	9 (33.3)	8 (32.0)
Duration of CNLBP [weeks]		96.0 (13.0; 1664.0)	96.0 (13.0; 1664.0)
Use of analgesic medication for CNLBP [n (%)]		15 (55.6)	13 (52.0)
Use of central muscle relaxants [n (%)]		4 (14.8)	4 (16.0)

Data are expressed as median (minimum; maximum) for continuous data and as absolute (relative frequency) for categorical data. HBRP: home-based rehabilitation programme; n: number of individuals; BMI: body mass index; CNLBP: chronic non-specific low back pain. Primary education: graduated from elementary school; secondary education: graduated from secondary school with state exam; tertiary education: graduated from university.

**Table 2 ijerph-20-01504-t002:** Adherence to home-based rehabilitation programme (patients who completed the HBRP).

	Patients Using Exercise Diary (n = 7)	Patients Using Mobile Application (n = 18)	Group Difference (*p*)
Percentage of completed records (%)	100 (100.0; 100.0)	97.1 (88.5; 98.9)	<0.001
Exercised at least once a day (%)	82.3 (80.1; 88.7)	94.8 (90.8; 97.9)	<0.001
Exercised twice a day (%)	72.9 (68.2; 78.4)	86.6 (75.3; 91.2)	<0.001
No exercise on the day (%)	17.7 (11.3; 19.9)	5.2 (2.1; 9.2)	<0.001

Data are expressed as median (minimum; maximum). Statistical comparison of values is performed using the Mann–Whitney test. n: number of individuals. The rehabilitation programme lasted 18 weeks (126 days) for all patients. That was the basis for the adherence calculation.

**Table 3 ijerph-20-01504-t003:** Patient satisfaction with the rehabilitation programme and change in patient beliefs—Results.

Question					
How satisfied are you with the completion of the rehabilitation programme?	Fully satisfied	Rather satisfied	Neither satisfied nor dissatisfied	Rather dissatisfied	Fully dissatisfied
	22 (88.0)	3 (12.0)	0 (0.0)	0 (0.0)	0 (0.0)
Do you believe that this rehabilitation programme has helped you with problems associated with your lumbar spine?	I believe it	I somewhat believe it	I don’t know	I somewhat don’t believe it	I don’t believe it at all
	20 (80.0)	4 (16.0)	1 (4.0)	0 (0.0)	0 (0.0)

Data are expressed as absolute (relative frequency).

**Table 4 ijerph-20-01504-t004:** Pain and disability—before and after rehabilitation programme.

Variable	Patients with CNLBP (before Rehabilitation Programme) (n = 25)	Patients with CNLBP (after Rehabilitation Programme) (n = 25)	Group Difference	
			Median (min; max)	*p*
Average pain intensity [NRS 0–10]	4.0 (1.0; 8.0)	2.0 (0.0; 5.0)	−2.0 (−4.0; 0.0)	<0.001
Current pain intensity [NRS 0–10]	2.0 (0.0; 6.0)	1.0 (0.0; 5.0)	−1.0 (−5.0; 1.0)	0.002
Maximum pain intensity [NRS 0–10]	7.0 (2.0; 10.0)	3.0 (0.0; 8.0)	−3.0 (−8.0; 4.0)	<0.001
Oswestry Disability Index [0–100%]	20.0 (4.0; 31.0)	4.0 (0.0; 15.5)	−13.0 (−24.4; −4.0)	<0.001
Roland–Morris Disability Questionnaire	7.0 (0.0; 19.0)	1.0 (0.0; 6.0)	−6.0 (−17.0; 0.0)	<0.001

Data are expressed as median (minimum; maximum). Statistical comparison of values is performed using the Wilcoxon test. n: number of individuals; NRS: Numerical Rating Scale; CNLBP: chronic non-specific low back pain.

**Table 5 ijerph-20-01504-t005:** Use of medication—before and after rehabilitation programme.

Variable	Patients with CNLBP (before Rehabilitation Programme) (n = 25)	Patients with CNLBP (after Rehabilitation Programme) (n = 25)	Group Difference (*p*)
Use of analgesic medication for CNLBP [n (%)]	13 (52.0)	3 (12.0)	0.006 ^†^
Use of central muscle relaxants [n (%)]	4 (16.0)	1 (4.0)	0.349 ^††^

Data are expressed as absolute (relative frequency). n: number of individuals; CNLBP: chronic non-specific low back pain. Statistical comparison of values is performed using Pearson’s chi-squared test with continuity correction ^†^ or Fisher’s Exact Test ^††^.

**Table 6 ijerph-20-01504-t006:** Functional parameters of trunk muscles—Before and after rehabilitation programme.

Variable		Patients with CNLBP (before Rehabilitation Programme) (n = 25)	Patients with CNLBP (after Rehabilitation Programme) (n = 25)	Group Difference	
				Median (min; max)	*p*
Maximal isometric lower back extensor strength	Prone position [kg] †	16.6 (4.3; 24.6)	23.4 (9.0; 34.1)	6.5 (0.5; 13.5)	<0.001
	Sitting position [kg] †	35.3 (13.5; 77.7)	57.7 (23.0; 86.3)	16.3 (−3.7; 37.2)	<0.001
	Standing position [kg] †	32.7 (9.7; 66.9)	51.8 (17.0; 71.8)	13.7 (−10.3; 41.9)	<0.001
Respiratory muscle strength	Maximum inspiratory pressure [cmH_2_O] ††	83.1 (27.3; 127.0)	91.7 (45.3; 138.7)	11.3 (−15.3; 61.3)	0.005
	Maximum expiratory pressure [cmH_2_O] ††	130.3 (52.0; 245.0)	147.7 (61.7; 267.3)	18.7 (−17.0; 98.3)	0.001
Trunk muscle endurance tests	Biering–Sørensen test [time (s)]	58.0 (4.0; 204.0)	122.0 (35.0; 247.0)	59.0 (−4.0; 177.0)	<0.001
	Prone-plank test [time (s)]	57.0 (5.0; 240.0)	83.0 (50.0; 212.0)	43.0 (−60.0; 112.0)	<0.001
	Side-bridge test-right side [time (s)]	25.0 (2.0; 122.0)	58.0 (16.0; 120.0)	35.0 (−2.0; 67.0)	<0.001
	Side-bridge test-left side [time (s)]	20.0 (2.0; 122.0)	61.0 (17.0; 121.0)	33.0 (−30.0; 69.0)	<0.001

Data are expressed as median (minimum; maximum). Statistical comparison of values is performed using the Wilcoxon test. n: number of individuals; CNLBP: chronic non-specific low back pain; MIP: maximum inspiratory pressure; MEP: maximum expiratory pressure. † Maximal isometric lower back extensor strength is calculated from the mean value of the 2nd to the 5th attempts. †† Respiratory muscle strength is calculated from the mean value of the 2nd to the 4th attempts.

## Data Availability

The data presented in this study are available on request from the corresponding author. The data are not publicly available.

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
