# Peer review of "Adherence and Effect of Home-Based Rehabilitation with Telemonitoring Support in Patients with Chronic Non-Specific Low Back Pain: A Pilot Study"

_ijerph, 2023, doi:10.3390/ijerph20021504_

Round 1

Reviewer 1 Report

Line 44: Low back pain (LBP)

NLBP, Chronic back pain, CNLBP, CLBP, LBP

These terms are used in your manuscript. Are they distinct? Or are you indicate the same thing? In order to define in detail the low back pain you are targeting, I recommend that you add a clear description so that these are distinguished.

Line 71: Is “back school” a general term? If not, I think it would be better to add a supplementary explanation about it or revise to something like Back School Exercise. Back school exercise

Line 71 and 84-86: These sentence state about Back School. I recommend that you reconsider sentence structure. Can you combine these sentences into one paragraph.

In addition, please consider whether these sentences are necessary for your background.

Line 106-108: This sentence indicates the methods and parameters. I think it would be better if it was described along with your research hypothesis.

Line 117-118, 130-132: Were these examinations performed in different procedures? Or is it a duplicate sentence?

Figure1 : why Why not a regular check up every 3 weeks?

Line 228-229: Proper paragraphing should be reconsidered.  

Table 1. You should state the characteristics of subject of 25, not 27.

Result 3.2 and Table 2. Can you show the participants' total number of exercise days (scheduled and actual)?  Were all subjects planning the same number of days, or were there differences among subjects? 

Line 261-264 and Table 3: Did you consider whether there were significant.

Table 4, 6: Please provide SDs.  

FIgure 3: Note that the upper right and lower left of the figure show the same numbers. I recommend that you remove the numbers in the bottom left row as they are duplicates.

Discussion: Please also provide a statement in terms of whether your hypothesis has been substantiated.

Author Response

Dear Editor,

Please find attached the revised version of the manuscript [IJERPH] Manuscript ID: ijerph-2076045, titled "Adherence and effect of home-based rehabilitation with telemonitoring support in patients with chronic non-specific low back pain: a pilot study " by Peter Krkoška and Daniela Vlažná for publication in the International Journal of Environmental Research and Public Health, in the Special Issue "Management of Patients with Chronic Diseases with Virtual Rehabilitation, Telerehabilitation and Remote Monitoring" as an original article. It has been amended in the light of reviewers´ comments. We found the comments highly valuable and tried to accommodate them to the highest possible degree. All the changes made in the manuscript are highlighted using “track changes”. We would be grateful if you could therefore consider publishing of the article.

Explanations and notes in response to the comments of the reviewer 1

1.Q: Line 44: Low back pain (LBP), NLBP, Chronic back pain, CNLBP, CLBP, LBP. These terms are used in your manuscript. Are they distinct? Or are you indicate the same thing? In order to define in detail the low back pain you are targeting, I recommend that you add a clear description so that these are distinguished.

1.A: These terms are not identical. Low back pain (LBP) is a superordinate term. Other terms describe the LBP type in more detail. The term chronic LBP (CLBP) indicates the duration of pain (more than 12 weeks) and non-specific LBP (NLBP) denotes pain that is not attributable to a recognisable, known, or specific pathology (e.g. tumour, osteoporosis, infection, fracture, structural deformity), radicular syndrome, or cauda equina syndrome. The term chronic non-specific low back pain (CNLBP) is a combination of the terms above. Terms and abbreviations are explained in the article (lines 39-47). Participants in our study suffered from CNLBP. The terms have been unified in the manuscript.

2.Q. Line 71: Is “back school” a general term? If not, I think it would be better to add a supplementary explanation about it or revise to something like Back School Exercise.

2.A. Back school is a general term. It describes an educational programme that includes information on the anatomy of the back, biomechanics, optimal posture, ergonomics, and back exercises and was designed to reduce back pain and prevent recurrences of LBP episodes. The definition of “back school” has been corrected in the manuscript.

3.Q. Line 71 and 84-86: These sentence state about Back School. I recommend that you reconsider sentence structure. Can you combine these sentences into one paragraph.

3.A. The proposed changes have been made.

4.Q. Line 106-108: This sentence indicates the methods and parameters. I think it would be better if it was described along with your research hypothesis.

4.A. The hypothesis of this study is that adherence and satisfaction with a home-based rehabilitation programme (HBRP) in patients with CNLBP will be favourable and after completing this programme, both patient-oriented parameters and the functional parameters of trunk muscles will improve. The aim of this study was to assess applicability of the HBRP with telemonitoring support and to evaluate adherence to exercise, patient satisfaction and the effect of this programme.

The text has been corrected and the hypothesis has been added to the introduction part of the manuscript.

5.Q. Line 117-118, 130-132: Were these examinations performed in different procedures? Or is it a duplicate sentence?

5.A. It was a duplicate sentence. It has been corrected in the manuscript. The chapters have been put in a more suitable order.

6.Q. Figure 1: Why not a regular check up every 3 weeks?

6.A. We have chosen to do a check-up two weeks after gaining a new exercise set as we wanted to be sure that all patients perform the exercise as correctly as possible. Letting the participants perform the exercise for three weeks without controlling the performance would risk an incorrect motor learning process.

7.Q. Line 228-229: Proper paragraphing should be reconsidered.  

7.A. The paragraph has been corrected in the manuscript.

8.Q. Table 1. You should state the characteristics of subject of 25, not 27.

8.A. Characteristics of subjects who completed HBRP (n=25) have been added to Table 1.

9.Q. Result 3.2 and Table 2. Can you show the participants' total number of exercise days (scheduled and actual)? Were all subjects planning the same number of days, or were there differences among subjects? 

9.A. The rehabilitation programme lasted 18 weeks (126 days) for all patients. That was the basis for the adherence calculation (Table 2). This explanation has been added as a comment to Table 2.

10.Q. Line 261-264 and Table 3: Did you consider whether there were significant.

10.A. The data express patient satisfaction and change in patient beliefs. The data show that most patients were fully satisfied with the HBRP (88%) and believe that this HBRP has helped them with problems associated with their lumbar spine (80%). We do not see the benefit in counting additional statistics as the result in this study group is obvious. A limitation for further statistical analyses is the relatively small number of patients in the study.

11.Q. Table 4, 6: Please provide SDs. 

11.A. The data don´t have normal distribution, therefore continuous variables were described using median, minimum, and maximum.  Using the mean and standard deviation would be incorrect.

12.Q. Figure 3: Note that the upper right and lower left of the figure show the same numbers. I recommend that you remove the numbers in the bottom left row as they are duplicates.

12.A. Figure 3 has been corrected.

13.Q. Discussion: Please also provide a statement in terms of whether your hypothesis has been substantiated.

13.A. The statement regarding our hypothesis has been added to the discussion part of the manuscript. The hypothesis of this study has been proved.

Kind regards,

Peter Krkoška

University Hospital Brno Jihlavska 20, Brno, 62500 Czech Republic

E-mail: krkoska.peter@fnbrno.cz

Tel.: 00420532232351

Reviewer 2 Report

This is a well-designed pilot study evaluating adherence to home-based exercise participation in people with CLBP utilizing an app or a diary. Secondary aims were to explore clinical and strength/endurance changes after the exercise intervention. Participants were highly adherent in both methods and there were functional and clinical benefits after the intervention. The background is well written, would consider combining paragraphs on motor control training to develop rationale. There are suggestions for completeness in methods and clarity and argument-strengthening in discussion in the comments section of attached PDF. I'd strongly suggest you create a supplementary data file including details on specific exercises included in books 1-4 with citations on where exercises were adapted from. I'd also suggest you look at CONSORT guidelines to be sure you include all relevant information on intervention. 

Author Response

Dear Editor,

Please find attached the revised version of the manuscript [IJERPH] Manuscript ID: ijerph-2076045, titled "Adherence and effect of home-based rehabilitation with telemonitoring support in patients with chronic non-specific low back pain: a pilot study " by Peter Krkoška and Daniela Vlažná for publication in the International Journal of Environmental Research and Public Health, in the Special Issue "Management of Patients with Chronic Diseases with Virtual Rehabilitation, Telerehabilitation and Remote Monitoring" as an original article. It has been amended in the light of reviewers´ comments. We found the comments highly valuable and tried to accommodate them to the highest possible degree. All the changes made in the manuscript are highlighted using “track changes”. We would be grateful if you could therefore consider publishing of the article.

Explanations and notes in response to the comments of the reviewer 2
Suggestions: I'd strongly suggest you create a supplementary data file including details on specific exercises included in books 1-4 with citations on where exercises were adapted from. I'd also suggest you look at CONSORT guidelines to be sure you include all relevant information on intervention.  
A: The booklets were created in the mother tongue of the participants. That is why we do not see a reason to publish them. We have added supplementary material that contains photographs of the exercise from our rehabilitation programme as they are depicted in the patient booklets. 
We checked the CONSORT guidelines; however, this is only partially applicable to our study because the guidelines are designed for RCTs.

1.Q: Line 71: consider removing this sentence and connecting with next paragraph on line 73 to build gap in one paragraph exercise intervention selection.
1.A: This line has been removed to line 85.

2.Q: Line 104: Consider using alternative word than 'effect' since this is not a randomized trial.
2.A: We reconsidered using another word besides “effect” but we were unable to find a better expression. 

3.Q: Line 165: Briefly comment on reliability of measures. State units of measure for outcome measures used.
3.A: The reliability of measures is clarified in our previous work. The citation following the sentence on which this comment was made (cit. number 28), is the primal original work, an observational study that aimed to define a battery of tests that comprehensively assess trunk muscle function (strength and muscular endurance) and to unify methodology in terms of consistency of measures (equipment and units). As the methodology of the assessment is quite extensive, as well as the methodology of the current intervention, we decided to arrange the methodology of this work by referring to the primal work and its deeply described methodology of assessment, so that the methodology of this work would not be overwhelming for the reader. The units used in particular measurements are presented in the tables along with the results.

4.Q: Line 178: Consider taking this piece out and creating another sentence describing back school intervention separately.
4.A: The definition of “back school” has been corrected in the introduction part of the manuscript. Supplementary material was added to the manuscript, where our rehabilitation programme, including back school, is briefly described.

5.Q: Line 184: Given this is a new intervention, briefly describe what clinical criteria were used for advancement.
5.A: The decision was made according to the quality and quantity in which the patient was able to perform the current exercise. If the patient was not able to follow the expected amount of exercise in the best quality, they would be asked to continue on the same level for a while. This information was added to the text. 

6.Q: Line 195: A summary of all the exercises in book 1 to 3? The last booklet had the most advanced exercises in it? Clarify.
6.A: The last booklet was handed to the patient at the final visit, when the intervention was over. It contained a summary of all exercises; it did not contain any new exercises, so the level did not increase further. The fourth booklet served to advise patients to continue with the exercises as instructed, the necessity of regular exercise in preventing the recurrence of back pain was explained. This clarification is a part of the text.

7.Q: Line 203: Confirm adherence reporting study instructions for both app and written diary. Daily adherence recording is more reliable than 1-week recall.
7.A: All patients were asked to report the performance of exercise immediately after the session was completed. The instruction was the same for both groups; however, it was only possible for us to check the group using the application. That is why all patients were advised to use the mobile application. Unfortunately, a few of them were not able to use modern technology, so we were not able to check their compliance immediately after the performance. The diary was handed in at every check-up visit and any problems that arose were discussed in person. (Line 221-222)

8.Q: Line 215: State rationale for median vs mean.
8.A: The data was non-parametric, that is why we used median.

9. Q: Line 241: Consider adding outcomes reported here for ease of reader -  median (min;max) etc..
9.A: All data are explained below the table, including this information in the table would make it unclear.

10. Q: Line 249: State in methods how patients were alocated to diary or app. It appears here as if patient choice was allowed.
10. A: Lines 213-218 in the methodological part and lines 366-368 in the discussion of the manuscript explain that for the purpose of this study a special mobile application was created. However, if the patient did not want/was not able to use this modern technology, there was an option to use a diary. This means that all patients were strongly encouraged to use the app, but they were not excluded from the study if they were unable to do so, and used a diary instead. The cohort was too small to allow us to exclude one-third of participants due to their lack of digital skills. It also shows us that digital literacy still cannot be taken for granted.

11. Q: Line 276: State post-intervention changes were measured on entire cohort somewhere.
11. A:  Lines 140-142 in the methodology part state that there was a post-intervention assessment after 18 weeks of HBRP.

12. Q: Line 360-363: Consider clarifying the point these sentences are trying to make, it is not obvious.
12. A: We believe that these sentences are understandable, proofreader did not indicate any problems with comprehension.

13. Q: Line 373-375: Consider stating how this study is similar or different to your study.
13. A: In the text we want to discuss possible aspects that may affect the adherence to home-based rehabilitation. We used this study for comparison as both our studies are made on patients with low back pain and the intervention is home based exercise.

14. Q: Line 376: Consider stating how this study is similar or different to your study.
14. A: We consider the context to be clear as we are compering how many participants dropped out of the mentioned study and our study. 

15. Q: Line 384-387: describe group qualitative study was done on for clarity.
15. A:  The following sentence explains the relation of the quotation to our study: All of these components were covered in the current study. (Line 407)

16. Q: Lines 397-400: Consider making statement why this research is relevant to your research. As well, if relevant state how current study findings shape future study design.
16. A: It is relevant as it follows a sentence claiming a positive accomplishment of our work – high satisfaction with the programme, which was measured at the end of the trial. We are trying to point out that studies which also measured the satisfaction after a longer period of time show different results. We did not observe long term changes, so we cannot agree or disagree, however we warn the reader that it should be taken into account that the high satisfaction that is often measured at the end of programme may decrease over a period of time. In the section “Limitation of the study” (Lines 501-510) we state that we did not follow the changes in our patients over a long-term period, however we explain that it is because this study is a pilot project that was followed by another study which also contains long-term assessments.
17. Q: Line 402: Consider adding statement at end of this paragraph on how your study intervention was clinically beneficial in both groups and why both groups improved framed by your adherence findings.
17. A: We believe that we did declare this point clearly in the same paragraph: This study showed a significant reduction in pain and a decrease in patient disability (ODI and RMQ). At the same time, it showed significant improvement in all functional parameters of the trunk muscles.

18. Q: Line 403-405: cite this
18. A: The sentence which follows the highlighted text still belongs to the indicated statement. This means that citation 37 belongs to both first sentences of the paragraph.

19. Q: Lines 417-420: This is interesting but how does it relate to your study findings?
19. A: We thought it would be interesting for readers to also discuss different aspects of other authors' research. We agree, that these results are redundant and we eliminated this part of the text from the manuscript. 
20. Q: Lines 421-427: I'd consider massively reworking this paragraph as arguments are not strong or cut it completely. The dosing argument is more complicated than frequency of exercise - it includes intensity (resistance level), frequency and duration. I do think a viable argument is you have good adherence for your home-based intervention and an intervention effectiveness trial is warranted as is exploring long-term effects on clinical outcomes per the second study you cite in this paragraph.
20. A: We agree with your comment, it is a very difficult question, there are different modalities which make the training effective. This is what we claim in the text, there is a need for more studies that would compare these two types of rehabilitation programmes, because we admit that the mentioned study, even though it has a completely different structure, has the same good results, (More studies are needed comparing different exercise intensities in terms of frequency as well as home-based rehabilitation with conventional rehabilitation in a medical setting to estimate and compare the size of the effect  of each programme. Lines 444-447).
21. Q: Lines 460-463: Consider clarifying this point further. Consider point that given there was no association between functional measures on patient-oriented outcomes, future studies should evaluate for additional drivers impacting patient experience (fear and catastrophizing being most common in PT literature).
21. A: Pain and disability are subjective parameters that reflect a number of psychosocial factors (e.g. patient's mood, fear, emotions, degree of pain catastrophizing).  Functional parameters could also be affected by pain, e.g. inability to perform a functional examination due to pain and not due to weakness.  In our study, patients improved in functional and patient-oriented parameters, but the relative changes did not correlate. So, the functional measures used in our study are not measures of pain-related behaviour. In future studies, including a bigger number of participants, different factors influencing outcomes should be analysed. To evaluate the effect of any intervention on CNLBP, both patient-oriented and functional outcomes should be considered (this is stated in our manuscript). 

We sincerely appreciate the possibility to submit our study and look forward to receiving any comments from you and your reviewers.

Please do not hesitate to contact us if you have any questions.

Kind regards,

Peter Krkoška

University Hospital Brno Jihlavska 20, Brno, 62500 Czech Republic

E-mail: krkoska.peter@fnbrno.cz

Tel.: 00420532232351

Reviewer 3 Report

Dear authors, I have read with interest the paper entitled "Adherence and effect of the home-based rehabilitation with telemonitoring support in patients with non-specific low back pain: a pilot study". I found the manuscript interesting and well written however it needs some revisions

1)In the introduction, in addition to reference 14, you could add: Scaturro D, Asaro C, Lauricella L, Tomasello S, Varrassi G, Letizia Mauro G. Combination of Rehabilitative Therapy with Ultramicronized Palmitoylethanolamide for Chronic Low Back Pain: An Observational Study. Pain Ther. 2020 Jun;9(1):319-326. doi: 10.1007/s40122-019-00140-9. Epub 2019 Dec 20.

2) Introduction check line 99

3)Line 104 the word "complete" is too generic and unscientific

4) In the introduction, describe only the goal, not the outcomes

5)In materials and methods you describe the back school method well

6) Check the table 5 and 6. Improve the figure 3 is poorly understood

7)In the discussion I advise you to consult:

Mbada CE, Olaoye MI, Dada OO, Ayanniyi O, Johnson OE, Odole AC, Ishaya GP, Omole OJ, Makinde MO. Comparative Efficacy of Clinic-Based and Telerehabilitation Application of Mckenzie Therapy in Chronic Low-Back Pain. Int J Telerehabil. 2019 Jun 12;11(1):41-58. doi: 10.5195/ijt.2019.6260. 

Peterson S. Telerehabilitation booster sessions and remote patient monitoring in the management of chronic low back pain: A case series. Physiother Theory Pract. 2018 May;34(5):393-402. doi: 10.1080/09593985.2017.1401190. Epub 2017

Author Response

Dear Editor,

Please find attached the revised version of the manuscript [IJERPH] Manuscript ID: ijerph-2076045, titled "Adherence and effect of home-based rehabilitation with telemonitoring support in patients with chronic non-specific low back pain: a pilot study " by Peter Krkoška and Daniela Vlažná for publication in the International Journal of Environmental Research and Public Health, in the Special Issue "Management of Patients with Chronic Diseases with Virtual Rehabilitation, Telerehabilitation and Remote Monitoring" as an original article. It has been amended in the light of reviewers´ comments. We found the comments highly valuable and tried to accommodate them to the highest possible degree. All the changes made in the manuscript are highlighted using “track changes”. We would be grateful if you could therefore consider publishing of the article.

Explanations and notes in response to the comments of the reviewer 3

1.Q.  Introduction check line 99
1.A. We've checked line 101, no error found.

2.Q. Line 104 the word "complete" is too generic and unscientific
2.A. The word “complete” is not in line 110, we do not understand the comment.

3.Q. In the introduction, describe only the goal, not the outcomes
3.A. We added a hypothesis and corrected the aims of the study in the introduction part of the manuscript.

4.Q.  In materials and methods you describe the back school method well
4.A. The definition of “back school” has been corrected in the introduction part of the manuscript. Supplementary material was added to the manuscript, where our rehabilitation programme, including back school, is briefly described.

5.Q. Check the table 5 and 6. Improve the figure 3 is poorly understood
5.A. Figure 3 has been corrected and we hope we made it more understandable. We have checked tables 5 and 6.

6.Q. In the discussion I advise you to consult:
Peterson S. Telerehabilitation booster sessions and remote patient monitoring in the management of chronic low back pain: A case series. Physiother Theory Pract. 2018 May;34(5):393-402. doi: 10.1080/09593985.2017.1401190. Epub 2017
Mbada CE, Olaoye MI, Dada OO, Ayanniyi O, Johnson OE, Odole AC, Ishaya GP, Omole OJ, Makinde MO. Comparative Efficacy of Clinic-Based and Telerehabilitation Application of Mckenzie Therapy in Chronic Low-Back Pain. Int J Telerehabil. 2019 Jun 12;11(1):41-58. doi: 10.5195/ijt.2019.6260.  
6. A.  We have added the citation Mbada et al., citation number 25.

We sincerely appreciate the possibility to submit our study and look forward to receiving any comments from you and your reviewers.
Please do not hesitate to contact us if you have any questions.

Kind regards,
Peter Krkoška
University Hospital Brno Jihlavska 20, Brno, 62500 Czech Republic
E-mail: krkoska.peter@fnbrno.cz 
Tel.: 00420532232351

Round 2

Reviewer 1 Report

I have reviewed all corrections and comments.

Thank you very much for your kind corrections and comments.

I look forward to continuing to develop your study.

Reviewer 3 Report

The manuscript has been improved and the authors have edited very well